Macular function measured by Nidek MP-3 microperimeter in normal Chinese individuals across age groups

Hu Dan
Wang Qiuhai
Liu Qingcui
http://orcid.org/0000-0002-0848-827X Pazo Emmanuel Eric
Liu Zhe
Wu Shuang
Zhang Wen
Xing Yan
Yu Bo yubo4950@tmu.edu.cn
Tianjin Key Laboratory of Retinal Functions and Diseases, Tianjin Branch of National Clinical Research Center for Ocular Disease, Eye Institute and School of Optometry, Tianjin Medical University Eye Hospital , Tianjin , China
Fogt Nick
Electronic publication date: 2025 Sep 9
Publication date: 2025
Volume: 13
Electronic Location ID: e19961
Received 2025 Mar 20; Accepted 2025 Jul 30
Copyright: © 2025 Hu et al.
Copyright year: 2025
Copyright holder: Hu et al.
License: This is an open access article distributed under the terms of the Creative Commons Attribution License, which permits unrestricted use, distribution, reproduction and adaptation in any medium and for any purpose provided that it is properly attributed. For attribution, the original author(s), title, publication source (PeerJ) and either DOI or URL of the article must be cited.
License URL: https://creativecommons.org/licenses/by/4.0/

Keywords: Microperimeter, Macular sensitivity, Fixation stability, Macular function

Funding: Tianjin Binhai New Area Health Research Project 2024BWKY18 Tianjin Medical University “Clinical Talent Training 123 Climbing Plan” Tianjin Medical University Eye Hospital High-level Innovative Talent Program YDYYRCXM-E2023-01 The Tianjin Key Medical Discipline (Specialty) Construction Project TJYXZDXK-037A This research was supported by the Tianjin Binhai New Area Health Research Project (2024BWKY18), Tianjin Medical University “Clinical Talent Training 123 Climbing Plan”, and Tianjin Medical University Eye Hospital High-level Innovative Talent Program (YDYYRCXM-E2023-01). The Tianjin Key Medical Discipline (Specialty) Construction Project (TJYXZDXK-037A) also provided support. The funders had no role in study design, data collection and analysis, decision to publish, or preparation of the manuscript.

==============================
Background

Microperimetry has emerged as a tool for assessing retinal function, especially macular function in recent years. In this study, the MP-3, a widely adopted device, was employed to analyze macular sensitivity and fixation stability across various age groups.

Methods

The research involved evaluating the macular sensitivity and fixation stability of 182 healthy eyes free from ocular disorders. The relationship between macular sensitivity, fixation pattern, and age was determined through generalized estimating equations.

Results

Macular sensitivity and percentage of fixation stability within 2° and 4° were measured for 182 eyes (mean age: 45.24 ± 17.03 years) using MP-3. The mean macular sensitivity was 29.5 ± 1.6, while the mean percentage of fixation points within a 2° circle was 93.0 ± 7.5%, and within a 4° circle was 98.3 ± 2.8%. Macular sensitivity, as well as the percentage of fixation points within the central 2° and 4° areas, tended diminish with advancing age.

Conclusion

In conclusion, among normal subjects, macular sensitivity and fixation values decrease with age. MP-3 emerges as a reliable instrument for measuring macular function.

Introduction

The functional evaluation of the macula is traditionally centered around measurements of visual acuity (VA). Nonetheless, there are instances where patients with normal VA continue to report visual symptoms, such as relative scotoma, metamorphopsia, and reduced contrast sensitivity (Sukha & Rubin, 2009; Rios et al., 2024). Given that VA does not fully capture the essence of functional vision, the advent of microperimetry is pivotal in the analysis of fundus-related retinal sensitivity. This technology is particularly crucial in the detection of macular dysfunctions that may not be apparent through conventional VA assessments alone (Horie et al., 2023; Pfau et al., 2021). Microperimetry stands out as an advanced method for assessing the functional integrity of the posterior pole across a spectrum of retinal conditions. This technique, which has been refined over the past decade, provides detailed, computerized mapping of the central visual field (Cassels et al., 2018; Iacono et al., 2019). Compared to standard automated perimetry, the MP-3 perimetry (NIDEK) is equipped with an auto-tracking system, which enables the measurement of visual fields in patients with paracentral or unsteady fixation (Ishiyama, Murata & Asaoka, 2015; Rodrigues Neto et al., 2023). Additionally, the sensitivity map produced by the perimetry exhibits an accurate point-to-point correlation with the retinal structure shown in the fundus photograph (Josan et al., 2023; Soans, Smith & Chung, 2025). The MP-3 microperimetry represents a significant advancement over the earlier MP-1 instrument, featuring an enhanced dynamic range of 0–34 dB. Additionally, the MP-3 offers a choice of two background luminance options (0.004 and 0.0314 cd/m2), whereas the MP-1 was limited to only one option, 0.004 cd/m2 background luminance (Balasubramanian et al., 2018). The maximum stimulus luminance of MP-3 is 10 cd/m2 compared with 0.4 cd/m2 of MP-1 (Liang et al., 2024; Georgiev et al., 2022). The increased stimulus luminance aids assessors in differentiating between relative and absolute scotoma (Taylor et al., 2022). The elevated background luminance prompts a greater involvement of cone cells in macular functional testing, thereby yielding more reliable results.

In an earlier study, macular sensitivity in a Western cohort of varying ages and genders was assessed using an MP-1 microperimeter (Sabates et al., 2011). A normative dataset for macular sensitivity using the MAIA microperimeter has been established in Switzerland (Pfau et al., 2024). Normative data for macular perimetry, obtained using the MP-3 microperimeter, have been established for a healthy Brazilian population (Rodrigues Neto et al., 2023). The MP-3 microperimeter was employed in a cohort of Chinese participants afflicted with macular disorders (Qian et al., 2022). However, the current literature on the use of the MP-3 microperimeter in healthy Chinese eyes is scarce. Consequently, we have designed the present study to ascertain the baseline levels of macular sensitivity and fixation condition among a Chinese cohort spanning various ages and genders. Our objective is to establish a normative database for the MP-3, which will assist in evaluating the extent of damage in patients with macular diseases across different age groups and genders.

Materials and Methods

Study design

This hospital-based, cross-sectional, observational study received ethical approval from the Ethics Committee of Tianjin Medical University Eye Hospital (Ethical Approval Number: 2017KY(L)-31). We recruited individuals who visited the ophthalmology clinic at our hospital with a diagnosis of asthenopia or ametropia. All participants provided written informed consent prior to their involvement in the study. A total of 97 participants were recruited. Each participant underwent ophthalmic examination, which included testing for best corrected visual acuity (BCVA), slit-lamp biomicroscopy, dilated fundus examination, and non-contact tonometry.

All participants enrolled in the study met the following criteria: (i) BCVA of at least 0.8; (ii) No history of eye diseases that could potentially affect retinal sensitivity, such as glaucoma, macular disorders, or optic nerve abnormalities; (iii) Measured refraction ranging from −6.00 to +3.00 diopters; (iv) Normal appearance of the optic nerve head; (v) Ability to cooperate well with the examiner during the testing procedure; (vi) Intraocular pressure (IOP) between 10 and 21 mmHg; (vii) No systemic diseases that might influence retinal sensitivity.

The sample size for the study was determined utilizing G*Power software. An effect size (q) of 0.5 was assumed, with a significance level (α error) set at 0.05 and a desired power of 0.95. This approach ensures a robust and statistically rigorous evaluation of the relationship between macular sensitivity, fixation stability, and age across various age groups.

Microperimetry measurement

The MP test was conducted after the enrolled eyes were dilated with 0.5% tropicamide, using the MP-3 microperimeter (NIDEK, Aichi, Japan). The machine’s parameters were set as follows: our protocol utilized 45 test points within an 8° circle centered on the macular lutea. The test employed a 4-2 staircase strategy at each point location, utilizing the Goldman size III stimulus. The background luminance was set at 0.0314 cd/m2, and the fixation target was a 1° diameter red circle. The stimulus dynamic range spanned 0 to 34 dB (Fig. 1). Prior to testing, operators provided a detailed explanation of the testing procedures and instructions on how to cooperate. All tests were administered by an experienced examiner. To mitigate the “fatigue effect,” which can diminish retinal sensitivity, patients were given a 2-min break following 5 min of continuous testing. Only reliable data, characterized by zero false positive and false negative rates with stable fixation, were collected for analysis.

Figure 1 Test parameters and representative results from MP-3 microperimeter.

The protocol employed 45 test points within an 8° circle centered on the macular fovea. The test was based on a 4–2 staircase strategy at each point location with the Goldman size III stimulus.

Statistical analysis

Statistical analysis was conducted using SPSS (version 27). Continuous data were expressed as mean ± standard deviation (SD). Generalized estimating equations (GEE) were employed to evaluate the correlation between parameter variables (macular sensitivity, fixation at 2°, and fixation at 4°) and age, assuming a Gaussian distribution and employing an exchangeable working correlation structure to account for within-subject correlation. GEE was also used to compare these outcome variables between the two gender groups, with statistical significance set at P < 0.05.

Results

Based on the specified parameters, the calculated sample size required for the study was determined to be 180 eyes. In accordance with the study’s inclusion criteria, out of the 194 eyes initially screened, 12 were excluded due to inadequate fixation or the presence of any false-negative or false-positive responses during the testing process. The characteristics of the final cohort of enrolled subjects are presented in Table 1. The average age of the participants was 45.2 ± 17.0 years, spanning an age range from 21 to 82 years. The mean macular sensitivity was measured at 29.5 ± 1.6 dB. Additionally, the mean percentage of fixation points within the circle at 2° was 93.0 ± 7.5%, and for the circle at 4°, it was 98.3 ± 2.8%.

Table 1 Demographic characteristics of subjects.

Sex, n	Male, 44 Famale, 47	
Age (years), n	21–40, 42	
41–60, 27	
>60, 22	
Laterality, n eyes	OD, 91	
OS, 91	
BCVA, n	0.8, 12	
1.0, 170	
Refractive error, n	+3D~0D, 104	
−0.25D~−3D, 66	
−3.25D~−6D, 12	
Note:

BCVA, best corrected visual acuity.

As depicted in Table 2, the mean macular sensitivity, the mean percentages of fixation points within 2°, and the mean percentages of fixation points within 4° circles for male participants were 29.4 ± 1.8 dB, 92.7 ± 8.3%, and 98.2 ± 3.0%, respectively. In contrast, for female participants, these values were 29.7 ± 1.5 dB, 93.3 ± 6.7%, and 98.5 ± 2.6%, respectively. No statistically significant differences were detected between the two gender groups (P > 0.05).

Table 2 Sex comparison of macular sensitivity and fixation distribution using generalized estimating equations.

Sex	Sensitivity	Fixation	
2°	4°	
Male	29.4 ± 1.8	92.7 ± 8.3	98.2 ± 3.0	
Female	29.7 ± 1.5	93.3 ± 6.7	98.5 ± 2.6	
β	0.232	0.572	0.260	
SE	0.305	1.127	0.441	
95% CI	[−0.365 to 0.829]	[−1.637 to 2.782]	[−0.603 to 1.123]	
P value	0.447	0.612	0.555	
Note:

Data are presented as means ± standard deviations (SDs), SE, Standard error; CI Confidence interval.

Table 3 displays the correlation between age and macular sensitivity, and the percentages of fixation points within 2° and 4° circles analyzed using GEE. Macular sensitivity, and the percentages of fixation points within both the 2° and 4° circles, negatively correlate with age. The normal values of macular sensitivity by age group are as follows: For individuals aged 20–39 years, the mean value is 30.4 dB with a 95% confidence interval (CI) ranging from 30.1 to 30.8 dB. For those aged 40–59 years, the mean value is 29.7 dB with a 95% CI of 29.4 to 30.1 dB. In the age group of 60 years and above, the mean value is 27.8 dB with a 95% CI of 27.3 to 28.2 dB.

Table 3 Generalized estimating equations analysis of the correlation between age and microperimeter measurement.

	β	SE	95% CI	P value	
Macular sensitivity	−0.062	0.007	[−0.075 to −0.048]	0.000*	
Fixation at 2°	−0.113	0.028	[−0.16 to −0.059]	0.000*	
Fixation at 4°	−0.033	0.011	[−0.055 to −0.010]	0.004*	
Notes:

* Statistically significant from generalized estimating equations.

SE, Standard error; CI, Confidence interval.

Discussion

Multiple studies have substantiated the clinical utility of the MP-3 microperimeter in assessing macular function across a range of retinal pathologies. These include central serous chorioretinopathy, glaucoma, age-related macular degeneration, and diabetic retinopathy (Cassels et al., 2018; Li et al., 2023; Phuljhele et al., 2021; Abreu-Gonzalez, Alonso-Plasencia & Gomez-Culebras, 2022). However, studies providing comprehensive reference values and normative data for macular sensitivity and fixation stability across different age groups remain relatively scarce. In this study, we evaluated macular sensitivity and fixation stability in healthy subjects using the MP-3, with a particular focus on age-related influences.

Upon analyzing the impact of age on macular sensitivity measurements, a negative correlation between age and sensitivity in normal subjects was observed. This finding aligns with the results of previous studies. A similar study that assessed macular sensitivity in healthy subjects using the MP-1 reported that a statistically significant decrease in sensitivity was exclusively evident in the age group of participants above 70 years (Sabates et al., 2011). Upon analyzing the discrepancy, we attributed it to the higher stimulus and background luminance of the MP-3 compared to the MP-1. This enhancement in luminance allows the MP-3 to provide more accurate and sensitive results, leading us to consider our findings as more reliable. In another study where MP-1 was used on 66 eyes, the conclusion was that there was a decline in macular sensitivity with increasing age in normal, healthy subjects (Shah & Chalam, 2009). In a study employing the MAIA microperimeter, another research team discovered a statistically significant inverse correlation between age and macular sensitivity. Similarly, other studies utilizing both MP-1 and MAIA microperimeters have identified declines in macular sensitivity with age, which aligns with our findings (Molina-Martin, Pinero & Perez-Cambrodi, 2017). The reduction in pupil size, age-related neural loss along the visual pathway from the retina to the brain, and the decreasing transparency of the ocular media have all been identified as contributing factors to the age-related decline in retinal sensitivity (Lachenmayr et al., 1994). Recently, a comparison of macular sensitivity values from the MAIA microperimeter and the Nidek MP-3 highlighted that both devices effectively detect retinal functional changes and scotomas. However, inter-device variability should be considered in future studies incorporating multiple devices (Marmalidou et al., 2025).

In terms of fixation stability, our findings revealed a negative correlation between age and the percentage of fixation points. This aligns with the findings of Molina-Martin, Pinero & Perez-Cambrodi (2017) who reported that the stability of the fixation pattern in healthy eyes tends to decrease with age when assessed using the MAIA microperimeter. Shah & Chalam (2009) observed a possible linear decline in fixation stability with age, a pattern similar to that of light sensitivity as measured by the MP-1. Forty participants were included in their study with the age range of 19–71 years. In contrast, Sabates et al. (2011) found no statistically significant variation in fixation stability at 2° among different age groups after examining 169 normal eyes using the MP-1. The criteria for subject enrollment and the machines used can contribute to the differences observed in the results. To facilitate comparison, we also summarized other microperimetry studies in healthy populations, as presented in Table 4.

Table 4 Summary of microperimetry studies in healthy individuals.

PMID	Year	Nation	Manufacturer model	Study design	Enrollment strategy	Sample size	Age range (years)	Measurements	Conclusion	
18849636	2008	USA	NIDEK MP-1	Prospective, Single center	Clinic-based	66	19–71	Macular sensitivity; fixation stability within 2° and 4°	Among normal healthy subjects, there was a linear decline in light sensitivity with increasing age	
20472294	2010	Italy	NIDEK MP-1	Prospective, Multicenter	Not mentioned	190	20–75	Light sensitivity from four macular sectors; repeatability	MP-1 allows for an accurate, repeatable, topographically specific examination of the threshold in four areas	
21358460	2011	USA	NIDEK MP-1	Prospective, Single center	Community-based	169	21–85	Subfield and mean macular sensitivity	Retinal sensitivity significantly decreased only in participants aged ≧70 years and in the peripheral macula of those aged ≧60 years	
28127734	2017	Spain	CenterVue MAIA	Prospective, Single center	Clinic-based	237	10–70	Average threshold; macular integrity; fixation indexes; horizontal and vertical axes of fixation	Retinal sensitivity and stability of fixation tends to decrease with age	
28495907	2018	USA	CenterVue MAIA, NIDEK MP-3	Prospective, Comparative, Single center	Not mentioned	31	22–43	The retinal sensitivity; its corresponding luminance and contrast	Retinal sensitivity is higher with MAIA, while luminance and contrast sensitivity are lower compared to MP-3	
39298722	2023	Brazil	NIDEK MP-3	Cross-sectional	Not mentioned	74	28–68	The retinal sensitivity; fixation stability values within 2° and 4°	The results of this study provide a normal and age-matched database of MP-3 microperimetry	
39422918	2024	Switzerland	CenterVue MAIA	Prospective, Multicenter	Clinic-based	531	16–85	Eccentricity from the fovea, overlap with the central fixation target, and eccentricity along the four principal meridians	A linear model incorporating age and eccentricity effectively explains normal variations in mesopic microperimetry.	

The MP-1 is a widely used NIDEK device in macular and glaucoma studies, providing richer data than visual acuity alone, but with limitations like a narrow dynamic range and lower stimulus luminance. The MP-3, an upgrade addressing these issues, offers improved stimulus luminance, background luminance, and a wider dynamic range, enhancing the evaluation of macular function (Palkovits et al., 2018). Current clinical tools, including fundus cameras, optical coherence tomography (OCT), and fundus fluorescein angiography (FFA), predominantly concentrate on identifying structural abnormalities of the retina. However, functional assessments, which are primarily based on visual acuity, are not adequate for detecting subtle macular dysfunctions and paracentral scotomas that can significantly impact visual perception (Parravano et al., 2022).

Consequently, visual acuity alone is inadequate for a comprehensive quantitative analysis of macular function (Von der Emde et al., 2021). Standard automated perimetry is commonly employed to evaluate visual field deficits. However, when assessing central retinal function, certain disadvantages are inevitable. In many macular diseases, central scotomas are detected using this method, but the results only provide a general correlation with retinal structure, making it challenging to pinpoint dysfunctional areas precisely on the fundus photograph. Consequently, standard automated perimetry is more suited for optic neuropathy rather than macular diseases. In contrast, microperimetry supplies sensitivity and fixation data that complement visual acuity in the assessment of macular function. It has gained extensive usage in macular disease studies to evaluate both macular sensitivity and fixation stability. For instance, microperimetry has received much attention for studies of non-neovascular age-related macular degeneration, as several recent trials have employed it as an outcome measure, and it can be tailored to correspond to the steep gradients of retinal neuron density within the human central retina (Chang et al., 2024). Robbie et al. (2018) utilized microperimetry to measure sensitivity and the percentage of fixation within a 4° circle in their investigation into the feasibility of a novel intraocular lens for patients suffering from macular diseases. MP-3 was used by Macular Telangiectasia Type 2-Phase 2 CNTF Research Group to assess retinal sensitivity in patients with macular telangiectasia (Chew et al., 2019). Donati et al. (2018) used sensitivity and fixation stability as secondary endpoints in their study of dexamethasone for the treatment of cystoid macular edema secondary to retinal vein occlusion.

The MP-3 microperimeter currently lacks a normative database for macular sensitivity and fixation stability, a deficiency that hinders clinicians from differentiating abnormalities from age-related alterations. Our study aims to address this gap by establishing a normative database for the MP-3 in a Chinese population. However, several limitations must be acknowledged. Firstly, the small sample size, restricted to participants from Tianjin, China and the lack of details on the enrollment strategy may limit the generalizability of our findings to other ethnic groups. To provide more robust and high-quality information, multicentric studies covering a wider range of populations are needed to reduce bias and enhance the generalizability of the results. Secondly, although previous studies, including those by Palkovits et al. (2018) have reported high test-retest reproducibility, we did not assess this aspect in our current study. Thirdly, we did not concurrently measure retinal structural changes, which restricts the analysis of the relationship between macular sensitivity and retinal thickness.

Conclusions

This study successfully established a normative database for MP-3 macular sensitivity and fixation stability within a Chinese population. Our findings indicate that macular sensitivity and fixation stability exhibit age-related changes.

Supplemental Information

Supplemental Information 1 Raw data.

Data was collected from the MP-3 micropetrmeter.

Supplemental Information 2 STROBE checklist.

Additional Information and Declarations

Competing Interests

The authors declare that they have no competing interests.

Author Contributions

Dan Hu conceived and designed the experiments, prepared figures and/or tables, and approved the final draft.

Qiuhai Wang conceived and designed the experiments, prepared figures and/or tables, and approved the final draft.

Qingcui Liu performed the experiments, prepared figures and/or tables, and approved the final draft.

Emmanuel Eric Pazo performed the experiments, prepared figures and/or tables, and approved the final draft.

Zhe Liu performed the experiments, authored or reviewed drafts of the article, and approved the final draft.

Shuang Wu analyzed the data, authored or reviewed drafts of the article, and approved the final draft.

Wen Zhang analyzed the data, authored or reviewed drafts of the article, and approved the final draft.

Yan Xing analyzed the data, authored or reviewed drafts of the article, and approved the final draft.

Bo Yu conceived and designed the experiments, prepared figures and/or tables, authored or reviewed drafts of the article, and approved the final draft.

Human Ethics

The following information was supplied relating to ethical approvals (i.e., approving body and any reference numbers):

This hospital-based, cross-sectional, observational study was approved by the Ethics Committee of Tianjin Medical University Eye Hospital (Ethical Approval Number: 2017KY(L)-31).

Data Availability

The following information was supplied regarding data availability:

Raw data is available in the Supplemental Files.

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
