# Peer review of "Macular function measured by Nidek MP-3 microperimeter in normal Chinese individuals across age groups"

_PeerJ, doi:10.7717/peerj.19961_

## Round 0.1 · original submission · Major Revisions

The primary concern is that the characteristics of the participant population is not well-described. Please ensure that there are details provided to address this.

Reviewer 1 ·

Basic reporting

-

Experimental design

-

Validity of the findings

-

Additional comments

Summary: The authors have conducted a study documenting macular sensitivity and fixation stability using the Nidek Microperimeter MP-3 in healthy Chinese populations. The work is relevant. I have some queries and comments that the authors should address.

Details:
- Title: Mention “Nidek MP-3 microperimeter” in the title.

- Table 1: Please provide more details about the demographics, such as visual acuity and refractive errors for each eye. This would be helpful since you are establishing a normative dataset.

- Table 1: The age distribution appears somewhat uneven. Regardless, it would be helpful if the authors could compute and provide the age-stratified normative sensitivity values, i.e., means/medians with 95% confidence intervals for each age bracket (20-29, 30-39, etc).

- Study Design, Line 68: There is no mention of power analysis or sample size calculation to justify the number of subjects.

- Statistical Analysis, Line 95: Both eyes from single participants have been analyzed without accounting for inter-eye correlation. Please refer to: Armstrong RA. Statistical guidelines for the analysis of data obtained from one or both eyes. Ophthalmic Physiol Opt 2013, 33, 7–14. doi: 10.1111/opo.12009

- Results, Line 120: “Conversely, age showed no significant correlation with the percentage of fixation points within the 4° circle (r = -0.199, P = 0.007).” However, the p-value shows significance. Please clarify.

- Discussion section: The clinical significance of the weak correlation between age and fixation stability is not thoroughly discussed.

- Discussion section: It would also be useful if the authors could comment briefly on the MCID (Minimal Clinically Important Difference), i.e., how much change in sensitivity and fixation stability would be clinically meaningful for clinicians using the Nidek MP-3 microperimeter based on these results.

- Additional and newer references to be added:

1. Introduction, Around Line 61 - The authors can mention that there are prior published studies of the use of MP-3 in Chinese participants. However, these were on macular disorders. Qian T, Xu X, Liu X, Yen M, Zhou H, Mao M, Cai H, Shen H, Xu X, Gong Y, Yu S. Efficacy of MP-3 microperimeter biofeedback fixation training for low vision rehabilitation in patients with maculopathy. BMC Ophthalmol. 2022 Apr 28;22(1):197. doi: 10.1186/s12886-022-02419-6

2. Introduction, Line 50 - structure-function correlations, for example "Unifying Structure and Function Towards a Comprehensive Macular Evaluation in Eye Disorders: A Multi-Modal Approach Using Microperimetry and Optical Coherence Tomography," in IEEE Transactions on Biomedical Engineering, vol. 72, no. 5, pp. 1572-1584, doi: 10.1109/TBME.2024.3513234.

3. Somewhere in the Discussion section - Recently, sensitivity values from the MAIA microperimeter and the Nidek MP-3 were compared: "Comparison Between MAIA and MP-3 In Healthy Subjects and Patients With Diabetes, Diabetic Retinopathy, and Age-Related Macular Degeneration". Invest Ophthalmol Vis Sci. 2025 Mar 3;66(3):59. doi: 10.1167/iovs.66.3.59

Reviewer 2 ·

Basic reporting

The manuscript is fairly well written, in the expected structure, although the use of specialized terms can be improved, as suggested. The authors provide raw data in supplementary tables.

p. 4 The mean macular sensitivity, the mean percentage of fixation points of the circle at 2° and 4° were 29.54±1.62, 93.00±7.53%, and 98.33±2.78%.

Seems like there should be 4 numbers reported (2 circles, 2 metrics). Do you mean “within the 2° and 4° circles”?

p. 4, mean age:

Also, give the range.

p. 6, l. 39, subtle metamorphopsia

"subtle" is not necessary.

p. 6, l. 42, subtle macular dysfunctions

Again, “subtle” is not necessary.

p. 7, l. 47 MP-3 perimetry

MP-3 perimeter - give manufacturer.

p. 7, l. 51, boasting an enhanced dynamic range of 0-34 dB.

Consider another word - “boasting” sounds like sales promotion literature.

p. 7, l. 53, (4 asb and 31.4 asb)

I see that the device reports light levels in apostilbs, which, to my knowledge, is an outdated unit of luminance. Could you please provide the equivalent values in candelas per square meter (cd/m²), the SI unit for luminance, as required by most major journals?

p. 7, l. 58, In an earlier study,

This literature on normative databases seems a bit thin. How was the literature searched?

p. 8, l. 86, on the macular fovea

This description is anatomically confusing. Do you mean the macula lutea (yellow spot), the foveal center (variably defined by OCT), or the preferred retinal locus of fixation, which is a functional measure, not anatomical? This discussion of different ways to center the visual map may be helpful [1].

p. 9, l. 107, The average macular sensitivity

I believe that 2 digits of precision is excessive, given what the device reports. Please check and see if 1 digit is adequate.

p. 10, l. 140, sensitivity threshold

In vision science, sensitivity and threshold are mathematically reciprocal concepts. Please amend this phrasing.

p. 11, l. 159, Offering a richer dataset than visual acuity alone, the MP-1 nonetheless exhibits certain limitations.

This information was provided in the Intro. Can it be condensed or eliminated here?

p. 12, l. 181, For instance …

Microperimetry has received much attention for studies of non-neovascular AMD, as several recent trials have employed it as an outcome measure, and it can be tailored to correspond to the steep gradients of retinal neuron density within the human central retina.
Chang DS, Callaway NF, Steffen V, Csaky K, Guymer RH, Birch DG, Patel PJ, Ip M, Gao SS, Briggs J, Honigberg L, Lai P, Ferrara D, Sepah YJ. Macular sensitivity endpoints in geographic atrophy: exploratory analysis of CHROMA and SPECTRI clinical trials. Ophthalmol Sci. 2024;4(1):100351. PMID 37869030
Another paper is online (https://doi.org/10.1016/j.xops.2025.100777). This literature should be mentioned.

Experimental design

The authors report visual sensitivity and fixation stability in healthy human eyes of different ages, as measured with an MP-3 microperimeter. Their goal was to develop a normative database as a reference for other studies of diseases affecting the human central retina. The sample size was good (182 eyes, weighted toward younger ages). They report comparable fixation stability across ages and a decline in mean sensitivity with age.

p. 7, l. 68, Study design

More information about the enrollment strategy, please, as readers from countries with different health systems will want to understand. Why are these people in the hospital? How is eye care delivered in this region? This will be particularly unfamiliar to readers accustomed to US health care. Of special concern, since sensitivity dropped in persons aged >60 years, how was lens opacity assessed? What was the best corrected photopic visual acuity for this patient group?

p. 8, l. 79, Normal appearance of the optic nerve head;

Assessed now - by biomicroscopy?

p. 8, l. 84, in a dimly lit room

The background illumination is essential for getting standardized measurements. This description seems very vague. Can more details be provided? What does the manufacturer of the device recommend for ambient lighting?

p. 8, l. 88, and the fixation target was a 1° diameter red circle

Circle or disk? The border of a disk is a circle.

p. 8, l. 93, Only reliable data, characterized by zero false positive and false negative rates with stable fixation, were collected for analysis.

What percent of testing data did not meet these criteria? Over what time periods were the rates determined?

p. 9, l. 97, strength of the association

Is this a standard categorization scheme? Seems a bit generous.

p. 9, l. 104, or a high frequency of false-negative or false-positive responses

Be specific about these cut-points.

Validity of the findings

A normative database for this newer device could be very useful, as the relatively new MP-3 overcomes the technical limitations of the older MP-1, as the authors point out. However, to be truly useful, the authors should provide more information about the patient population from which the data were gathered, as there may be inherent biases. A single-site center always has limited generalizability, but more details about that single site will guide others who obtain results from different clinic populations.

p. 10, l. 129, Upon analyzing the impact of age on macular sensitivity measurements, a strong negative correlation between age and sensitivity in normal subjects was observed. This finding partially aligns with the results of previous studies.

More information is needed about the lens status of the participants before this age change is fully appreciated. This entire paragraph could consider the age structure and enrollment strategy of the participants in this study. A comparative table of literature might be useful.

p. 10, l. 141, linear declines in macular sensitivity with age, which aligns with our findings(20).

Did the authors consider other fits? The decline looks steeper after age 60 years.

p. 11, l. 151, The discrepancies between our results and those of Shah et al.

Can the authors be more specific, beyond the fact that different devices were used? What about the age structure of the study cohort and the enrollment strategies? Are they clinic-based or population-based? If clinic-based, what kind of clinic?

p. 12, l. 191. However, several limitations must be acknowledged. Firstly, the small sample size, restricted to participants from Tianjin, China, limits the generalizability of our findings to other ethnic groups.

The generalizability is also limited by the lack of details on the enrollment strategy, and this can be addressed.

---

## Round 0.2 · Minor Revisions

Thank you for your responses to the reviewer comments. Reviewer #1 has some remaining comments that will need to be addressed, primarily around the statistical analyses.

Reviewer 1 ·

Basic reporting

no comment

Experimental design

no comment

Validity of the findings

See additional comments

Additional comments

The authors have addressed most of my comments. However, there are still points that need to be addressed.

1. Statistical Analysis, Line 95: Both eyes from single participants have been analyzed without accounting for inter-eye correlation. Please refer to: Armstrong RA. Statistical guidelines for the analysis of data obtained from one or both eyes. Ophthalmic Physiol Opt 2013, 33, 7–14. doi: 10.1111/opo.12009
- Results, Line 120: “Conversely, age showed no significant correlation with the percentage of fixation points within the 4° circle (r = -0.199, P = 0.007).” However, the p-value shows significance. Please clarify.
Response: We appreciate your insightful suggestion regarding the statistical analysis and Results. we have revisited our statistical analysis in light of the need to account for the potential influence of both eyes of the same patient being included in the study. To address this, we have transitioned from a Pearson correlation test to a more robust Generalized Estimating Equations (GEE) approach. This method is better suited for analyzing clustered data, such as when both eyes of individuals are part of the dataset. Consequently, we have removed the previous interpretations based on the Pearson correlation test, as they are no longer applicable. This revision ensures a more scientifically rigorous and convincing analysis of the data. GEE results were shown in Table 3.

Updated Reviewer Comment - GEE is appropriate. However, please provide more details about the GEE implementation in the Statistical Analysis section. Which distribution was assumed? Eg: Gaussian? What was the “working correlation structure” - eg: exchangeable. Also in Table 3, please report the standard error values next to the beta values.

2. Statistical Analysis & Table 2: “To determine the differences in parameter variables between the two gender groups, the Student’s t-test was employed…”
Here, male vs female eyes are compared, treating the 182 eyes as independent. The authors can use the same GEE technique to account for the correlation between both eyes.

3. Other minor issues:
- Table 1: 95% CI is mentioned, but the values are missing
- Results: Paragraph about Table 2. “P>0.5” should be “P>0.05”

Reviewer 2 ·

Basic reporting

The authors have been highly responsive to reviewer comments and provided clearer descriptions of their work. They also incorporated additional literature.

Experimental design

The basics of enrollment strategy are clearer now. The authors reanalyzed data to account for two eyes from one patient, as requested by the other reviewer.

Validity of the findings

The findings are increased in validity because of the clearer description of enrollment strategy and the revised statistical analysis. It's still a single center study but with a clear description, others can build on the results.

Additional comments

No further comments from me. I commend the authors for their responsiveness. The manuscript is strengthened.

---

## Round 0.3 · accepted · Accept

Thank you for your responses to the latest reviewer comments.